# Evaluation of Accelerator Pedal Strength under Critical Loads Using the Finite Element Method

Kostyantyn Holenko [1], Eugeniusz Koda [2], Ivan Kernytskyy [2], Oleg Babak [1], Orest Horbay [3], Vitalii Popovych [3], Marek Chalecki [2,*], Aleksandra Leśniewska [2], Serhii Berezovetskyi [4] and Ruslan Humeniuk [4]

[1] Department of Tribology, Automobiles and Materials Science, Khmelnytskyi National University, 29016 Khmelnytskyi, Ukraine; holenkoke@khmnu.edu.ua (K.H.); babakop@khmnu.edu.ua (O.B.)

[2] Institute of Civil Engineering, Warsaw University of Life Sciences (SGGW), 02-787 Warsaw, Poland; eugeniusz_koda@sggw.edu.pl (E.K.); ivankernytskyy@ukr.net (I.K.); aleksandra_lesniewska@sggw.edu.pl (A.L.)

[3] Department of Equipment Design and Operation, Institute of Mechanical Engineering and Transport, Lviv Polytechnic National University, 79013 Lviv, Ukraine; orest.z.horbai@lpnu.ua (O.H.); vitalii.v.popovych@lpnu.ua (V.P.)

[4] Department of Mechanical Engineering, Lviv National University of Nature Management, 80831 Dublany, Ukraine; siko@email.ua (S.B.); ruslan.video@gmail.com (R.H.)

* Correspondence: marek_chalecki@sggw.edu.pl

**Abstract:** The core idea of the research consists in a formulation of boundary conditions of a mechanical accelerator pedal's strength in an Ansys environment, whose conditions are equivalent to full-scale tests under the critical loads defined by the UNECE's Regulation No. 13. The lack of regulatory requirements for the strength of pedal types other than brake pedals is a major gap in vehicle certification, especially when it comes to agricultural machinery. In such cases, the authors suggest being guided by UNECE R 13 regarding the strength of the accelerator and other types of pedals and checking their behavior under loads of at least 1000 N. The real value of the yield strength of the material (Silumin 4000) is very important, both in the physical real-life experiments and in FEA simulation. The critical case of a short-term shock loading of the pedal in its extreme position has been considered separately. With the help of the Ansys Explicit Dynamics module, results of a pedal's behavior were obtained; it lost its integrity and suffered destruction. It is also necessary to check the intermediate stress values depending on the loads for direct and hybrid tasks using the Transient Structural module in order to estimate other critical cases of the pedal behavior.

**Keywords:** strength analysis; tractor floor pedal; UNECE Regulations; FEA; Ansys; safety factor; Newton–Raphson method; Johnson–Cook equation



## 1. Introduction

The object of our research is an acceleration pedal assembly, model EAAH-MFP02-2340, used in vehicles, in particular in tractors (for example, model K 744R, etc.). A strength analysis of the floor pedal was proposed to be carried out in stages, with an assessment of the results of various loading modes (determining boundary conditions) that are maximally permissible in terms of the loads but unlikely in real operating conditions. According to the conditions of certification tests (UNECE Regulation No. 13), the load generated by the vehicle operator on the pedal should not be higher than 1000 N, which has been determined by the requirements of workplace ergonomics and comfort. In case the value of 1000 N is insufficient to turn on the system, then further tests are not carried out [1], because it is considered that the pedal design is ineffective (not able to ensure full transfer of loads to the control elements). The load application time defined by the UNECE Regulations has a decisive influence on the strength and integrity of the pedal; therefore, shock load calculations are extremely important when designing and certifying pedals of various

types. Hence, the main task of the research presented below is to formulate a methodology for assessing the compliance of the mechanical pedal strength with the requirements of UNECE R 13; our formulation is aided by a computer simulation method.

Maximum loads on mechanical pedals are regulated by UNECE R 13-09-2002, the only technical prescriptions for the official approval of road vehicles of categories M, N, and O regarding braking [1]. However, in the presented investigations, they have been applied to the accelerator pedal, which is more complex in terms of loads. The authors present a study of pedal performance characteristics and driving load on the slopes of a mountain road, referring to naturalistic driving in-work tests [2], which is especially relevant for analysis in conjunction with the UNECE R 35-00:2002 (Uniform technical prescriptions for official approval of road vehicles in relation to the placement of control pedals) [3], because workplace ergonomics and load requirements for the control elements are inherent during the operation of vehicles. The topic of the vehicle pedal has been taken up by Chen et al. [4], wherein the comfort of the driver's leg was assessed on the basis of biomechanics principles. Unlike UNECE R 13, the requirements of UNECE R 35 concern the ergonomics of the driver's workplace, compliance with demands for comfort, and efficiency of the control elements' interactions with the driver, which has been addressed in [3,4]. The study being presented is focused on investigations of the mechanical strength of pedals, which is rarely presented in modern publications. The study of the dynamic behavior of a driver coupled with a vehicle was presented by Reddy et al. [5], who described how the reaction during a pedal actuation impacts a human body state. This publication correlates with the present research in terms of the speed of the load application on the pedal, which is an important factor in the context of strength analysis (according to the requirements of UNECE R 13, the maximum load on the brake should not exceed 1000 N, but the speed of its application significantly affects the strength). Du et al. [6] and Xi et al. [7] simulated an interaction between a driver and a seat, focusing on the use of car pedals and the characteristics of the legs of elderly drivers, and relating this directly to ergonomics. Rantaharju et al. [8] presented health risks related to whole-body vibration and shock investigations, logically supplementing the previously cited works, and compared alternative assessment methods for high-acceleration events in vehicles. Since the highest loads on the pedals are observed precisely on brake pedals, the testing and analysis of car brake pedals with reduced weight, as well as the measurement and quality control of brake pedals, play an important role in the automotive industry (cf. Ergenc et al. [9], Cerilles et al. [10]). Noh et al. [11] examined automobile brake pedals, performing a load analysis and strength tests. These studies on load analysis and the durability test conditions of automobile brake pedal stress can be considered very close to the topic raised in the presented publication. Dhande et al. [12] presented a conceptual design and profile of the brake pedal.

Taking into account that the physical properties of the material define the safety factors of the pedal and its cracking, it is reasonable to mention the papers dedicated to the rupture of pedals. Setién et al. [13] studied cracking problems arising in the manufacturing of steel brake pedals by press forming. Cracks were detected in critical zones of the part, revealing a possible lack of ductility in the constitutive material. In order to overcome this problem, a detailed study of the behavior of the steel used has been carried out. In this way, a mechanical characterization, as well as a microstructural analysis, were performed of both cracked and uncracked pedals. Hfayedh et al. [14] addressed the mechanical properties and fatigue performance of materials such as aluminum alloy, which is important for the present research, as the pedal casting is manufactured using Silumin 4000. Shock loading and the materials' dynamic response were experimentally analyzed using Doppler velocimetry in order to provide adequate data for the simulation. The development of an automotive braking performance analysis program considering dynamic characteristics was discovered and reported in [15].

Among the theoretical aspects of the present study, explicit dynamic numerical schemes for solving any hyperbolic or parabolic nonlinear systems of partial differential equations (explicit loads, fluids, etc.), based on the Courant–Friedrichs–Lewy (CFL)

condition, and other factors, are important [16]. The Ansys Explicit Dynamics environment was used in the presented research. The Ansys Explicit Dynamics module is used for short-time event analysis (less than 0.5 s), nonlinearity processes with large deformations, solid/beam models with nonlinear material properties, contact-intensive loads, and other transient cases with plastic deformation (strain), and is based on Johnson–Cook equation [17]. The Johnson–Cook equation parameters have been described and explained in [14] as well.

It is quite challenging to estimate the braking performance of a vehicle because the brake system consists of many parts, including a booster, master cylinder, and caliper. Calculation of characteristics such as braking force in vehicle tests requires much time and money. Therefore, the development of a method of estimation of the braking performance of a vehicle using qualitative methods is beneficial. Jung et al. [18] presented a program that can analyze the braking capabilities of a vehicle, such as pressure, efficiency, and pedal travel. A reliability-based and deterministic design optimization of an FSAE pedal was presented in [19], along with the pedal design. Finally, Balakrishnan et al. [20] and Khandani [21] presented formulas dedicated to FEA, and a convergence of the Newton–Raphson method. These are important due to the type of calculations used in the study being presented, and have been provided in Section 2.3.

The topic of the pedal design is relevant for the present research because the configurations of castings, the cross-section profiles of the component parts, and the geometry of the pedal movement, all while taking into account the pedal's travel limitation mechanism, are part of this research, and all the listed factors affect the final strength of the pedal. In addition, it should be recalled that the requirements of UNECE R 13 apply to the brake pedal, but there are no certification rules that would apply separately to other types of pedals (accelerator, power take-off, clutch, etc.). At the same time, the issue of durability is relevant for accelerator pedals, because, in agricultural machines and tractors, the transmission of force from the pedal to the executive unit is provided by a remote control cable (push–pull cable), which can pass through a complex route and create significant resistance due to internal friction, and the executive units have their own operational specifics. Thus, in real operating conditions, significantly higher loads may occur on the mechanical pedal than on the electronic one, which was determined during experimental tests of the K 744R tractor model. Based on this, the novelty and relevance of the present research lie in the formation of an analytical FEA-based methodology for assessing the strength of the accelerator pedal and its compliance with the requirements of UNECE R 13, which is completely new and unexplored today in the field of the automobile/agricultural branch.

Having the above in mind, a research task addressed in this study was to develop a methodology for assessing the strength of other types of pedals than just brake pedals, which are governed by the requirements of UNECE R 13-09-2002.

## 2. Materials and Methods

The research object was the accelerator pedal assembly, model EAAH-MFP02-2340, which is used in vehicles, in particular in tractors (model K 744R, etc.) and mounted to the floor (Figure 1a). A simulation of full-scale tests based on FEA was performed in the Ansys Workbench 19.0-2022 R2 environment.

The calculation's solid model was prepared in the SolidWorks environment (STEP format), which accurately corresponded to the research object in terms of dimensions (Figure 1b), mass characteristics forming the model assembly, and the material characteristics (strength, yield point, Poisson's ratio, etc.), which were applied in Ansys. The same model was used in the tractor's manufacturer's assembly, so the accuracy of this solid model is maximal (without the simplifications). The FEA mesh depended on the calculation module used in Ansys: a pedal model in "Transient Structural" differed from that in "Explicit Dynamics" by the number of elements, etc. For example, the total number of finite elements was 17,881 and 37,034 nodes in the case of the "Direct task" described

below ("Transient Structural" module). The foot part of the pedal assembly was made of Silumin 4000, whose properties are presented in Table 1.

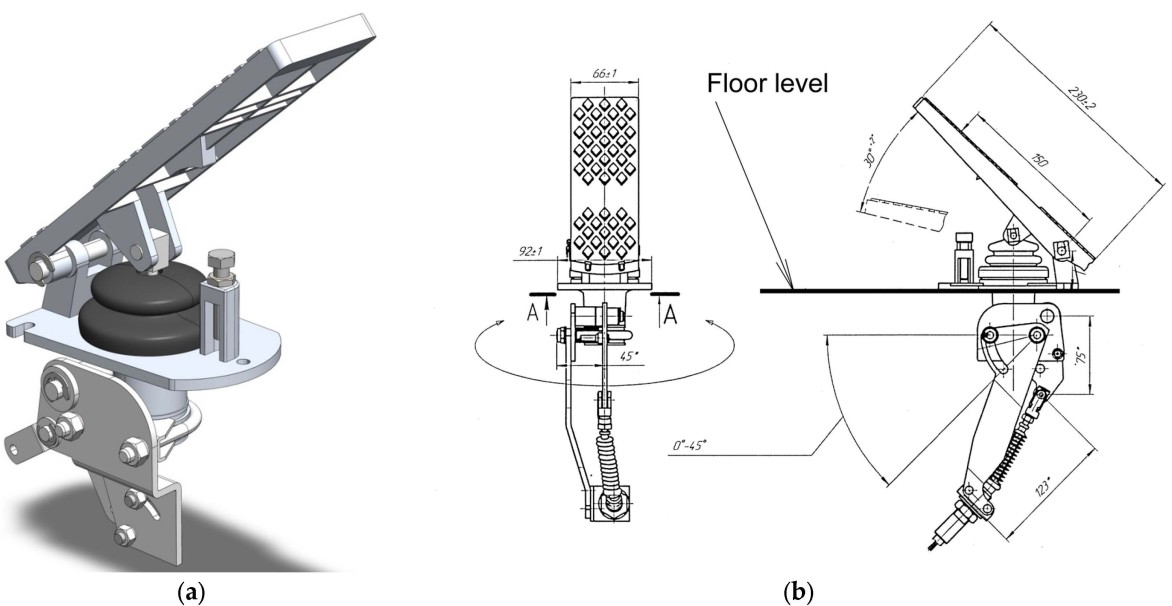

**Figure 1.** EAAH-MFP02 floor pedal: (**a**) 3D model in the Ansys Workbench environment—isometry; (**b**) assembly drawing of the pedal.

**Table 1.** Physical and mechanical properties of Silumin 4000 (EN AC-44400-F, AlSi9).

| | |
|---|---|
| Density | 2650 kg/m$^3$ |
| Poisson's ratio | 0.33 |
| Young's modulus | 71 GPa |
| Yield strength | 120–180 MPa |
| Tensile strength | 210 MPa |

An appearance of a solid model in the Ansys Workbench R19 software environment and a drawing of the pedal are presented in Figure 1.

The investigations encompassed four tasks.

### 2.1. Direct Task

The purpose of the direct task was to provide an analysis of the behavior of the solid model under a direct load transferred from a driver's leg through the foot part of the pedal, rod, and system of levers to the executive mechanism (via the control push–pull cable). According to a number of identical UNECE requirements [1,3], the maximum permissible load on the pedal by an operator is 1 kN. This force was applied uniformly, normally to the surface of the foot part of the pedal (tag A—Figure 2a), in the Transient Structural module interface (Figure 2), which allowed us to track the model behavior at any moment of the time interval.

The dynamics of the load application in the full-scale test conditions was simulated using the loading steps presented in Figure 3a.

Obviously, the model required an application of displacement constraints, otherwise it would not be statically balanced. The restrictions were set according to the actual fixing of the pedal in the vehicle (Figure 4): (a) rigid fixing (fixed support) in the faces of the mounting holes for the bolted connections of the pedal base with the vehicle floor frame (tag A—Figure 4); (b) limitation of vertical displacements (along the *Y* axis) of the pedal base using the "Displacement" type constraint in Ansys (part B—Figure 4). Displacement limitation (b) means the mounting of the pedal on the floor surface (floor level is shown in Figure 1b). Horizontal displacements along the *X*, *Z* axes were allowed. In addition,

a Cylindrical Support-type restriction of the displacements was applied in the eye of the push–pull cable armature (tag A—Figure 5).

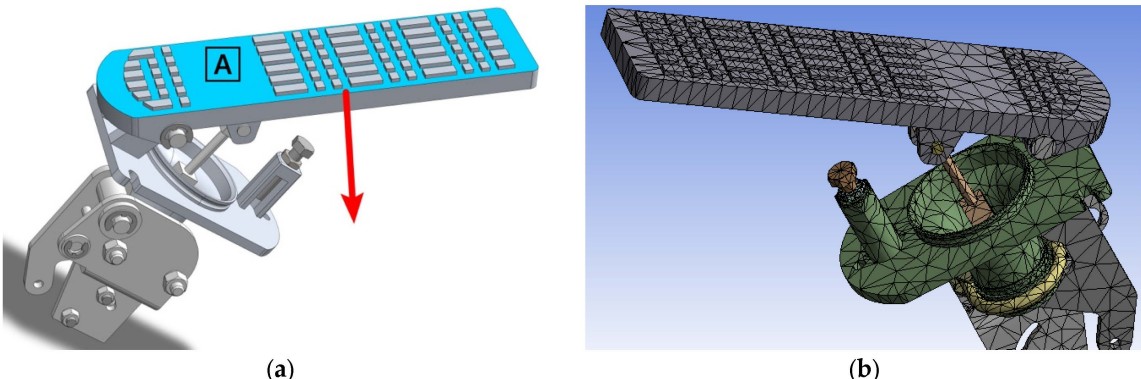

**Figure 2.** Foot pedal model: (**a**) scheme of applying load (red arrow) to the foot part of pedal; (**b**) FEA model.

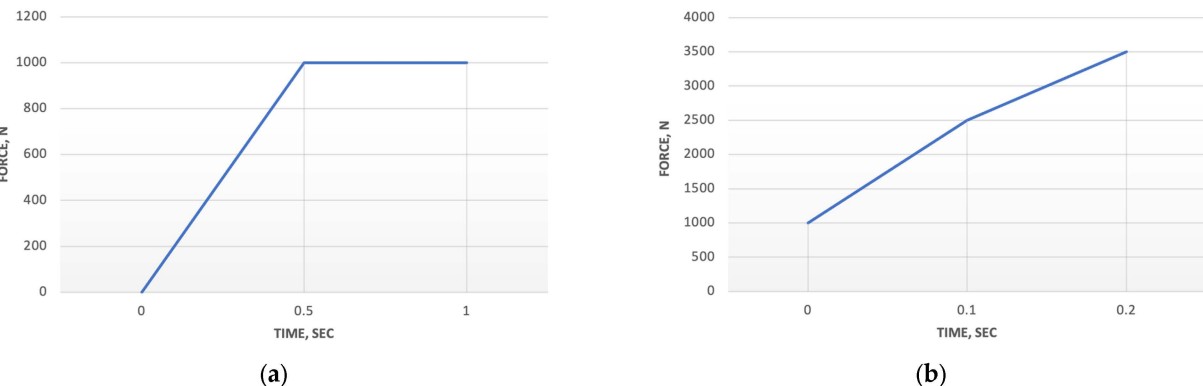

**Figure 3.** Load distribution in time: (**a**) direct task (Ansys Transient Structural); (**b**) shock load task (Ansys Explicit Dynamics).

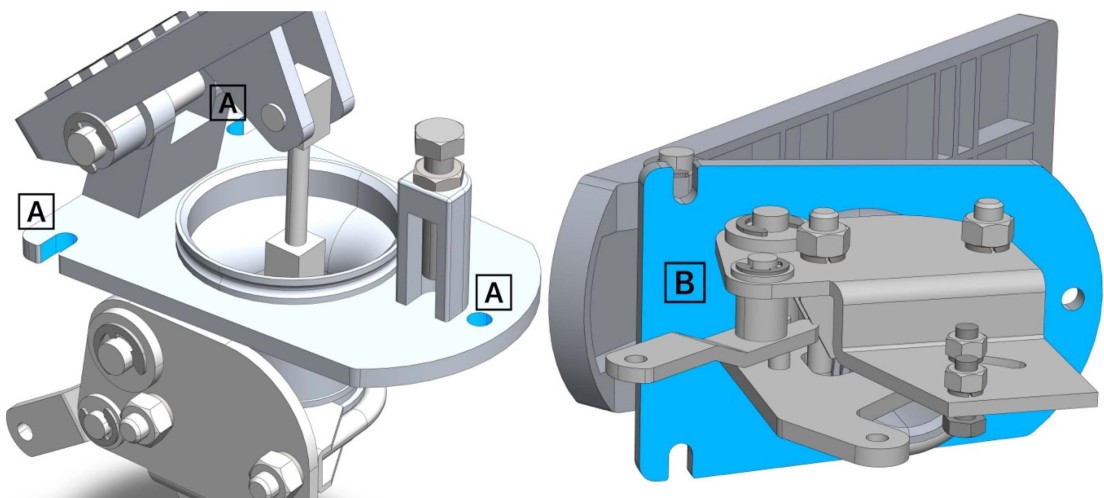

**Figure 4.** Constraints applied to a pedal model in Ansys Transient Structural analysis.

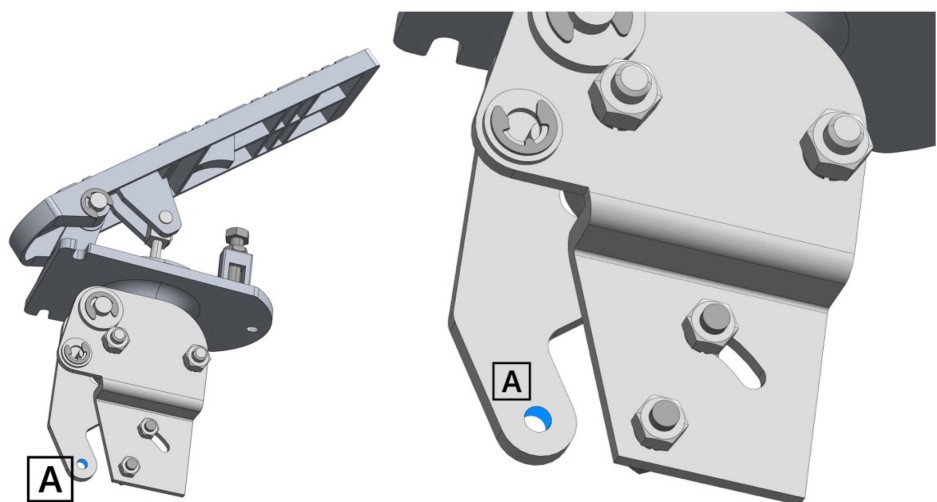

**Figure 5.** Limitation of Cylindrical Support-type displacements in the eye of cable fastening lever.

The pedal model was divided into its finite elements (Figure 2b). The total number of finite elements was 17,881; the model contained 37,034 nodes.

Ansys Transient Structural analysis is a prerequisite to ensure compliance with the requirements of UNECE R 13 regarding the stepwise application of the load up to its maximum value and its retention period. In this way, the simulation of full-scale tests in the FEA environment was performed. The use of the Transient Structural module had an additional benefit from the point of view of determining the uniform strength of the structure; during the loading process, we could control the stresses in the required locations while loading. Considering that the pedal was connected to the remote control cable, which itself had inertia, the application of Ansys Transient Structural analysis was relevant in this case as well (it was especially so for the hybrid task shown below).

### 2.2. Inverse Task

The inverse case required imposing opposite conditions, as load was transferred in reverse order: from the cable drive mounting lever (part A—Figure 6) to the foot pedal part (tag B—Figure 6).

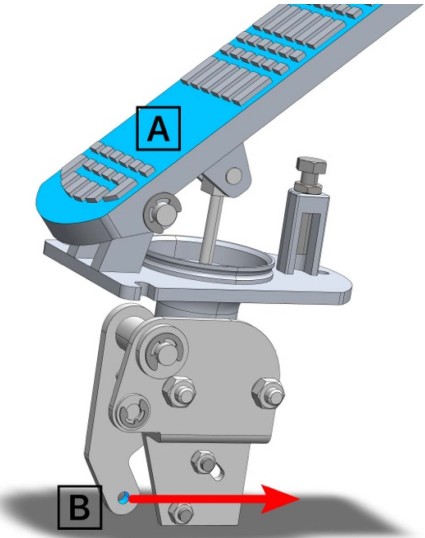

**Figure 6.** Pedal model: load for the inverse task (red arrow).

At the same time, constraints described in the direct task were preserved, and the movement of the foot part of pedal was limited by the fixed support of the area of its upper part (part A—Figure 6). Thus, the foot part of the pedal was deprived of the movement/rotation and would limit the travel of the push–pull cable connected to the mounting hole (tag B—Figure 6). The value of the force was 1000 N (it corresponded to the maximum load transmitted by the remote control cable), and its vector was directed along the X axis. The area of the force application corresponded to the cylindrical surface of the fastening fixing hole.

### 2.3. Hybrid Task

The most difficult, so-called symmetrical, case was analyzed within this task. The simultaneous application of opposite forces on the foot part of the pedal and on the fixing hole (Figure 7) of 1000 N was applied in area A (the upper part of the pedal in Figure 7), whereas the reaction was measured in the fixing hole (B in Figure 7). The reaction $R$, resulting from the following values of component forces: $F_x$ = 552.35 N, $F_y$ = 1928 N, had the value $R$ = 2005.5 N. The reaction along the $Z$ axis can be ignored (the module was incomparably small).

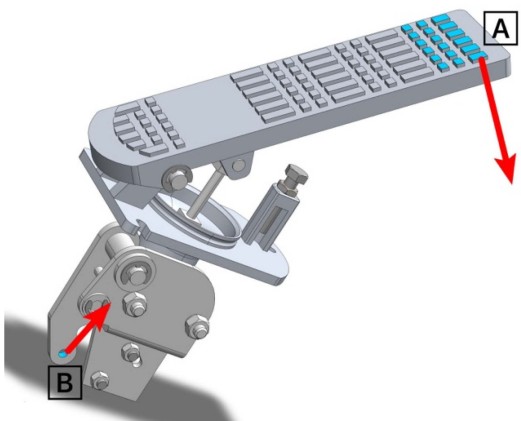

**Figure 7.** Load for the hybrid task (red arrows).

### 2.4. Shock Load Task

Before proceeding to the study of shock loads, the concepts of implicit and explicit types of calculations used in Ansys will be presented, and the theory of calculations used in the finite element method (FEM) will be investigated. In the general case, any static FEM analysis (for example, Static Structural in Ansys) is determined by a strict linear Equation (1), in which the concept of time dependence is absent (static process, but not transient) [20]:

$$[K]\{x\} = \{F\}. \tag{1}$$

A so-called transient or modal analysis (dynamic case) follows a more complex basis equation in the following form [20]:

$$[K]\{x\} + [C]\{\dot{x}\} + [M]\{\ddot{x}\} = \{F\}, \tag{2}$$

where:

[M], [C], [K]—mass matrix, damping matrix, and stiffness matrix, respectively;
$\{\ddot{x}\}$, $\{\dot{x}\}$, $\{x\}$—acceleration, velocity, and displacement, respectively;
{F}—load vector.

In the aim to find {x}, the matrix [K] needed to be inverted, which was not easy. The matrix [K] was unknown due to displacements of adjacent nodes; the convergence must be sought via iterations. In most cases, the Newton–Raphson method is used as the most suitable one because of its powerful potential for solving equations in a numerical way. The

core idea is based on the approach suggesting that a continuous and differentiable function can be approximated by a straight-line tangent to it. The Newton–Raphson method is a root-finding algorithm aimed at finding out the value of $x$ for which a function $f(x) = 0$. From a geometrical point of view, it can be treated as the value of $x$ where the function being sought crosses the X axis. The Newton–Raphson method is a technique used to find the roots of nonlinear algebraic equations. The basic formulation of the method is [21]:

$$x_{n+1} = x_n - \frac{f(x_n)}{f'(x_n)}. \tag{3}$$

where: $x_n$—a current estimation, $x_{n+1}$—a next estimation. Such an approach could be used to describe the displacements of adjacent nodes of the FEA model to find their position at any moment of time with the movement according to the nonlinear equation $f(x)$.

An implicit solution includes a "step by step" approach: current values at a current time step are provided by the values calculated at a previous time step. This case is also called the Euler time integration scheme. Opposite to the implicit approach, in the explicit analysis, the vector $\{\ddot{x}\}$ is sought instead of $\{x\}$, and the complex stiffness matrix inversion is bypassed in this way, so it is only needed to invert the mass matrix $[M]$.

The explicit scheme has a higher conditional stability than the implicit scheme, which is unconditionally stable for large time steps. However, an explicit calculation is stable if the time step size is smaller than the critical time step size of the investigated model; otherwise, the final results reached in the solution may be misleading. Integration schemes in the explicit approach use the central difference quotient to find out the velocities and accelerations at the current time step $t_n$, and then find out the unknown displacements at the next time steps $t_{n+1}$:

$$\dot{x} = \frac{x(t+\Delta t) - x(t-\Delta t)}{2\Delta t}; \ \ddot{x} = \frac{x(t+\Delta t) - 2x(t) + x(t-\Delta t)}{2\Delta t^2}. \tag{4}$$

where: $\{\ddot{x}\}$, $\{\dot{x}\}$, $\{x\}$—acceleration, velocity, and displacement, respectively; $\Delta t$—time step.

The time step for explicit calculations is based on the appropriate characteristic length of the finite element $L_e$ and the wave propagation speed $c$ in the medium or material. The element characteristic length is the shortest distance in the finite element through which a wave of stress can pass, so the time step scale factor (TSSF) in the Ansys Explicit Dynamics module was selected as 0.9 and used to ensure a step lower than the critical time step [20]:

$$c = \left( \frac{E(1-\nu)}{\rho(1+\nu)(1-2\nu)} \right)^{\frac{1}{2}}, \tag{5}$$

where: $c$—wave propagation speed, $\nu$—Poisson's ratio; $\rho$—density, $E$—Young's modulus.

The characteristic length $L_e$ was calculated based on the element geometric properties and for an octahedron (a typical kind of FE for a solid bodies grid), it could be found as [20]:

$$L_e = \frac{V}{A_{max}}. \tag{6}$$

where: $V$—the element volume, $A_{max}$—the area of the largest side.

The Ansys Explicit Dynamics module was used for the short-time event analysis (less than 0.5 s), the nonlinearity processes with large deformations, the solid/beam models with nonlinear material properties, contact-intensive loads, and other transient cases with plastic deformation (strain), based on Johnson–Cook equation [17]:

$$\sigma_y\left(\varepsilon_p, \dot{\varepsilon}_p, T\right) = \left[ A + B(\varepsilon_p)^n \right] \left[ 1 + C\ln\left(\dot{\varepsilon}_p^*\right) \right] \left[ 1 - (T^*)^m \right], \tag{7}$$

where: $\varepsilon_p$—equivalent plastic strain, $\dot{\varepsilon}_p$—equivalent normalized plastic strain rate (deformation speed), $A$, $B$, $C$—material constants, $n$—a mechanical hardening exponential;

*m*—thermal softening exponential. The normalized strain rate and a homologous temperature $T^*$ in the equation were defined as below [17]:

$$\dot{\varepsilon_p}^* = \frac{\dot{\varepsilon_p}}{\dot{\varepsilon}_{p0}}; \quad T^* = \frac{(T - T_0)}{(T_m - T_0)} \tag{8}$$

where the subscripts 0 represent initial values (room temperature), $\dot{\varepsilon_p}^*$—effective total strain rate normalized by quasi-static threshold rate (the strain rate being non-dimensionalized by the reference strain rate), $\dot{\varepsilon}_{p0}^*$—reference strain rate, $T$—current specimen temperature, $T_m$—fusion temperature. The expression in the third set of brackets (7) represents thermal softening such that the yield stress drops to zero at the melting temperature $T_m$.

An explicit dynamic numerical scheme for solving any hyperbolic or parabolic non-linear system of partial differential equations (explicit loads, fluids, etc.) can be stable and converge to the correct solution only if it satisfies the Courant–Friedrichs–Lewy (CFL) condition, which states that the full numerical domain of dependence must contain the physical domain of dependence (14). Actually, the CFL condition provides an upper bound for the local time step for a given element in an explicit numerical Equation (9). In the basic approach, the one-dimensional case is characterized by the following continuous-time model equation:

$$C_n = \frac{u \cdot \Delta t}{\Delta x} \leq C_{nmax} \tag{9}$$

where: $C_n$—Courant number; $u$—the magnitude of the velocity [m/s]; $\Delta t$—time step [s]; $\Delta x$—characteristic size of the mesh cell in one dimension [m]; $C_{nmax}$—a value changing with the method used to solve the discretized equation, especially depending on whether the method is implicit or explicit. In the case of an application of an explicit (time-marching) solver, for the most part, $C_{max} = 1$. The implicit (matrix) solvers are usually less sensitive to numerical instability; hence, larger values of $C_{max}$ may be tolerated as well. The CFL condition for the two-dimensional $(X - Y)$ and $n$-dimensional case in common is:

$$C = \frac{u_x \Delta t}{\Delta x} + \frac{u_y \Delta t}{\Delta y} \leq C_{max} \text{ or } C = \Delta t \left( \sum_{i=0}^{n} \frac{u_{x_i}}{\Delta x_i} \right) \leq C_{max}. \tag{10}$$

Finally, in order to understand the processes occurring during a material rupture in the Explicit Dynamics environment, the law of conservation of energy, on which the work of the specified Ansys module is based, should be considered. In this case, the energy error equation was defined in Explicit Dynamics as follows:

$$Energy\ Error = \frac{|CE - RE - WD|}{\max(|CE|, |RE|, |KE|)} \tag{11}$$

where: $E$—current energy (at the moment of calculation); $RE$—reference energy; $KE$—kinetic energy; $WD$—work done, which could be calculated as

$$WD = CE + L + BF + EE + PF \tag{12}$$

where: $CE$—work done by constraints; $L$—work done by loads; $BF$—work done by body forces; $EE$—energy removed by element erosion; $PF$—work done by contact penalty forces. The Explicit Dynamics "Energy Summary" graph is presented in Figure 8 and explained below (it is a screenshot of a graph made by Ansys as a part of the results).

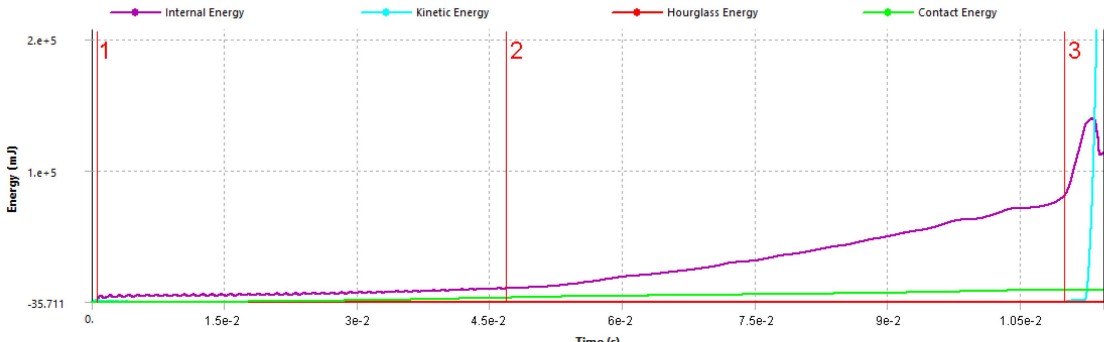

**Figure 8.** Explicit Dynamics energy conservation graph (lines 1, 2, 3—see description in text).

Thus, thanks to the above equations that systematize knowledge on the theory of the FEM approach used in Ansys, as well as being aware of the difference between implicit and explicit types of calculation (Equations (1)–(10)), it is easier to decide which Ansys calculation type is better for a particular task—in the case being presented, it was Ansys Explicit Dynamics. Understanding the essence of energy transformations in the Ansys environment and generally knowing the types of energies present in the process of pedal deformation (Equation (12)), an engineer or scientist can judge the graph of energy transformations (Figure 8). The process defined on the graph (Figure 8), divided by the red vertical lines, goes as follows:

- Before approaching line 1, the pedal has not yet made contact with the limiter (appropriate bolt regulating the maximum displacement of the pedal);
- Between lines 1 and 2, there is a contact of the pedal with the bolt, which is accompanied by a micro-vibration, which continues until the contact is stabilized (wavy line—"Internal energy");
- Between lines 2 and 3, the contact of the pedal behind the bolt stabilizes and the stage of elastic deformations begins as the load from the pedal increases;
- Between lines 2 and 3, short-term plastic deformations are observed (the graph of "Internal energy" goes up), as well as a material rupture (the "Kinetic energy" line rapidly rises, which means the separation of a broken part of the pedal and its further movement). This stage can be further observed in Figure 13.

An inquiry into the theory of explicit calculations in Ansys enabled us to investigate the critical case, when the load is applied to the pedal during the shortest possible time period and the pedal itself is located in the position before the contact (Figure 9): it rests against the limiter (appropriate bolt regulating the maximum displacement of the pedal—red arrow in Figure 9). Mesh quality control (Table 2) was used for the case of the Explicit Dynamics analysis model (Figure 9); this mesh was different from that used in the Transient Structural case.

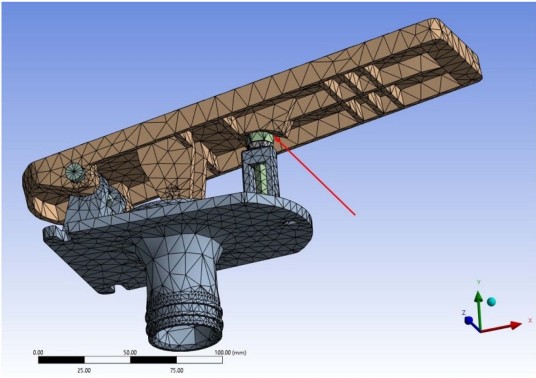

**Figure 9.** Pedal shock load calculation: solid model with boundary conditions.

**Table 2.** Reaction along the axes in fixing hole for the control cable connection.

| Quality Criterion | Warning (Target) Limit | Error (Failure) Limit | % Warning | # Warning | % Failed | #Failed | Worst |
|---|---|---|---|---|---|---|---|
| Min Element Quality | Default (0.2) | Default ($5 \times 10^{-4}$) | 0.179% | 23 | 0% | 0 | 0.155 |
| Max Aspect Ratio (Explicit) | Default (5) | Default (1000) | 0.529% | 68 | 0% | 0 | 6.744 |
| Min Characteristic Length (LS-DYNA) | Default (0.78 mm) | Default (0.078 mm) | 0.327% | 42 | 0% | 0 | 0.559 mm |
| Min Tet Collapse | Default (0.1) | Default ($1 \times 10^{-3}$) | 0.000% | 0 | 0% | 0 | 0.165 |

The system activation time corresponded to the time of pressing the pedal—0.2 s, according to the requirements of UNECE Regulation No. 13 [1] for tests. Thus, the task consisted in an analysis of the strength of the pedal model beyond the permissible limits of 1000 N under an instantaneous load of up to 3500 N for 0.2 s (Figure 3), in the Ansys Explicit Dynamics environment.

## 3. Results

### 3.1. Direct Task

It was proposed that we evaluate the research results on the basis of maximum stress of the investigated model (Figure 10a). Thus, the maximum stress value is 206 MPa and was observed in the bending area of the cable drive mounting lever, which corresponds to the cantilever type of load. The obtained result is within the material strength of this lever. For the pedal itself, however, the values of the maximum stresses were definitely lower (141 MPa) than the yield point of the material used to make the foot part of pedal (EN AC-44400-F, AlSi9)—180 MPa (Figure 10b).

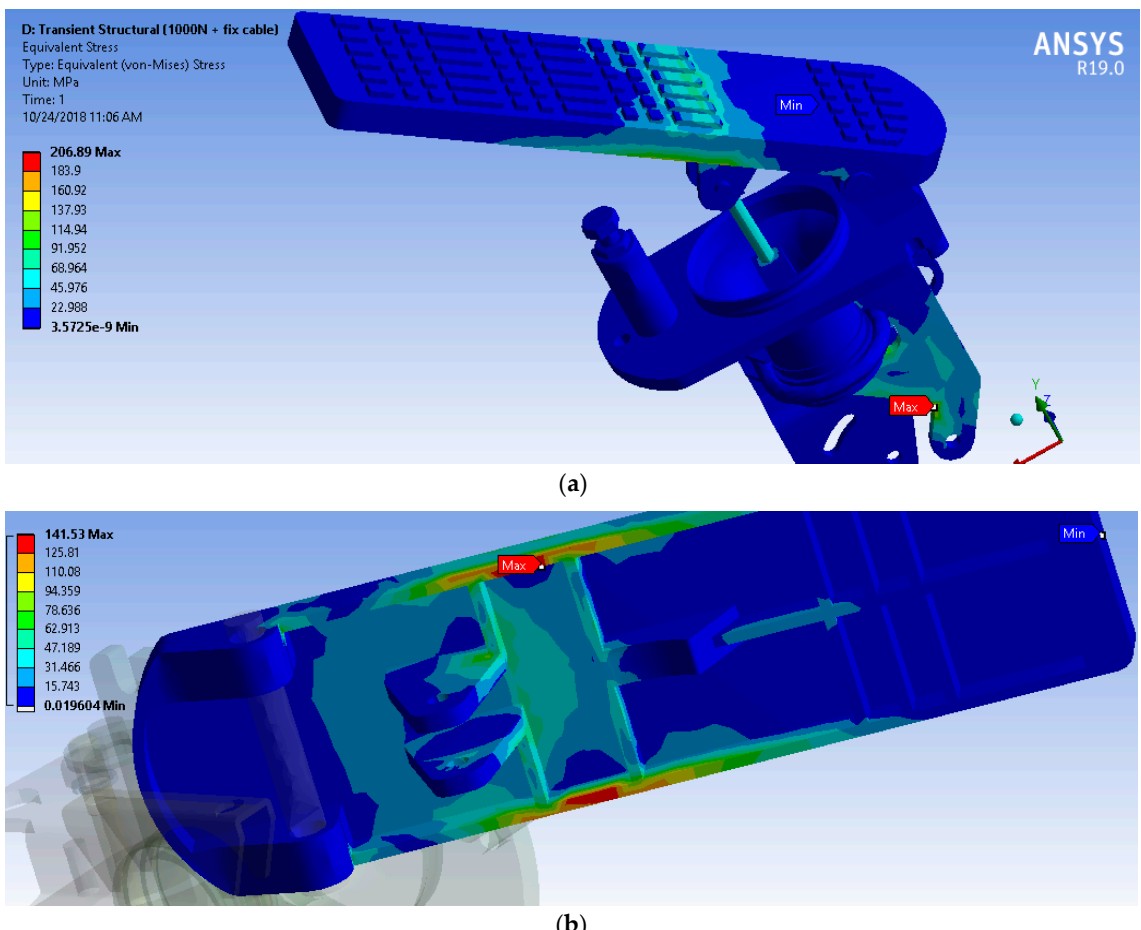

(**a**)

(**b**)

**Figure 10.** Stress map of the pedal assembly (direct task conditions): (**a**) upper side, (**b**) lower side.

### 3.2. Inverse Task

The load distribution character corresponded to that in the direct task shown above. The maximum stress was 37.6 MPa (Figure 11), which indicates a significant margin of strength.

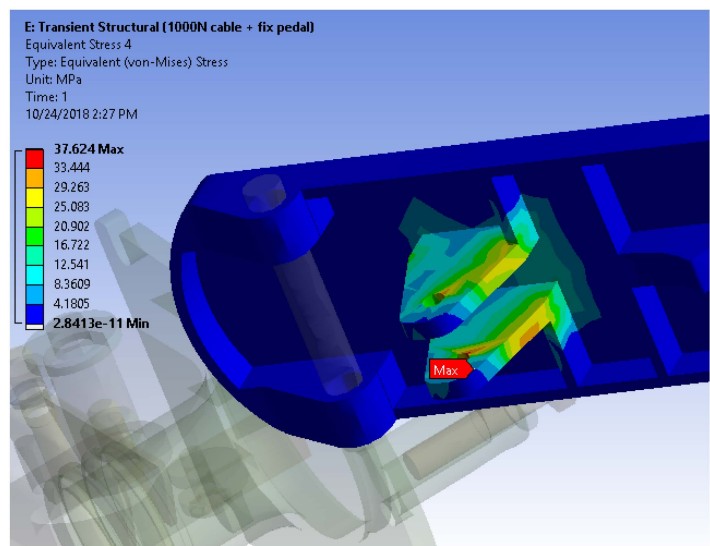

**Figure 11.** Stress map of the pedal's foot part (inverse task conditions).

### 3.3. Hybrid Task

The obtained results presented in the stress map (Figure 12a)—554 MPa—are beyond the strength of most steels, not to mention the yield point. Formally, this was indicated by a rupture of the material of the pin at the rotation axis of the foot pedal part (Figure 12b). The mentioned stress was found in the area of attachment to the indicated pin (Figure 12c). Taking into account the known yield strength of the material (EN AC-44400-F, AlSi9), equal to ca. 180 MPa, it can be stated that the full-scale sample of the tested model would not withstand the loads applied in this calculation mode; the destruction of the pedal foot part is possible in the area between the two pins (Figure 12c), as shown in the photo (Figure 1b).

### 3.4. Shock Load Task

The permissible yield strength, equal to 120 MPa for this most difficult case, corresponded to the lower strength limit of Silumin (casting under pressure). The time distribution of the dynamic load in the conditions of the full-scale tests is presented in Figure 3b: the loads started with 1000 N and continued up to 3500 N until the end of the period of 0.2 s. To perform this FEA, the following working station configuration was used: two physical Intel Xeon 24-core processors, RAM 48 Gb, NVIDIA GeForce 4 Gb video; total calculation time was 7 h 47 min.

The results of the shock loads task are as follows: being a relatively fragile material, Silumin showed a fracture in the appropriate location—the area of the pedal support which abuts against the bolt travel limiter (Figure 9). The stress was 124 MPa (Figure 13), which exceeded the yield point of the used material under the shock load conditions (when the load reached a value of ca. 2500–2600 N). The Ansys limit of the yield strength was set at 120 MPa for Silumin 4000 (Table 1). Silumin 4000 is a fragile material, as was evidenced in the real conditions of what was observed during the real tests (Figure 16). Although it demonstrated minimal plastic deformation during the load process, the pedal cracked almost at once as the load reached the maximum value (2255.5 N—Table 3), without demonstration of any visually significant plastic deformations and without reaching the 210 MPa of the tensile strength indicated by the manufacturer of Silumin 4000. It is visible on the video timer (Figure 16b)—both states of the pedal (intact and broken) occur in the same second of the video (00:36 s)—and this correlates with the FEA results

(Figure 13a—before the rupture and Figure 13b—at the first moment of rupture). This real-life behavior prompted the authors to set the stress–strain characteristics in "Plasticity toolbox" in the "Engineering data component" of the Ansys materials section as close to the pedal behavior of the natural sample of Silumin as possible. Thus, it became clear that it was necessary to focus on achieving the yield point, because the real value of the tensile strength of the pedal material was different than 210 MPa (Table 1). Moreover, as a result of testing the strength of full-scale pedal samples, the values of breaking loads ranged from 8 to 12%, which indicated differences in the technology of manufacturing castings. It was concluded, then, that the maximum value of 180 MPa of the yield strength of Silumin 4000 may be relevant for castings made under very high pressure. On the other hand, all of the pedal samples actually provided a twice-higher strength safety factor relative to the load of 1000 N required in UNECE R 13. The top part of the pedal cracked (Figure 14), so the structure has lost its integrity and cannot be used anymore. This was confirmed by the natural test results (Figure 14a)—photos of the destroyed samples demonstrate the rupture of the pedal.

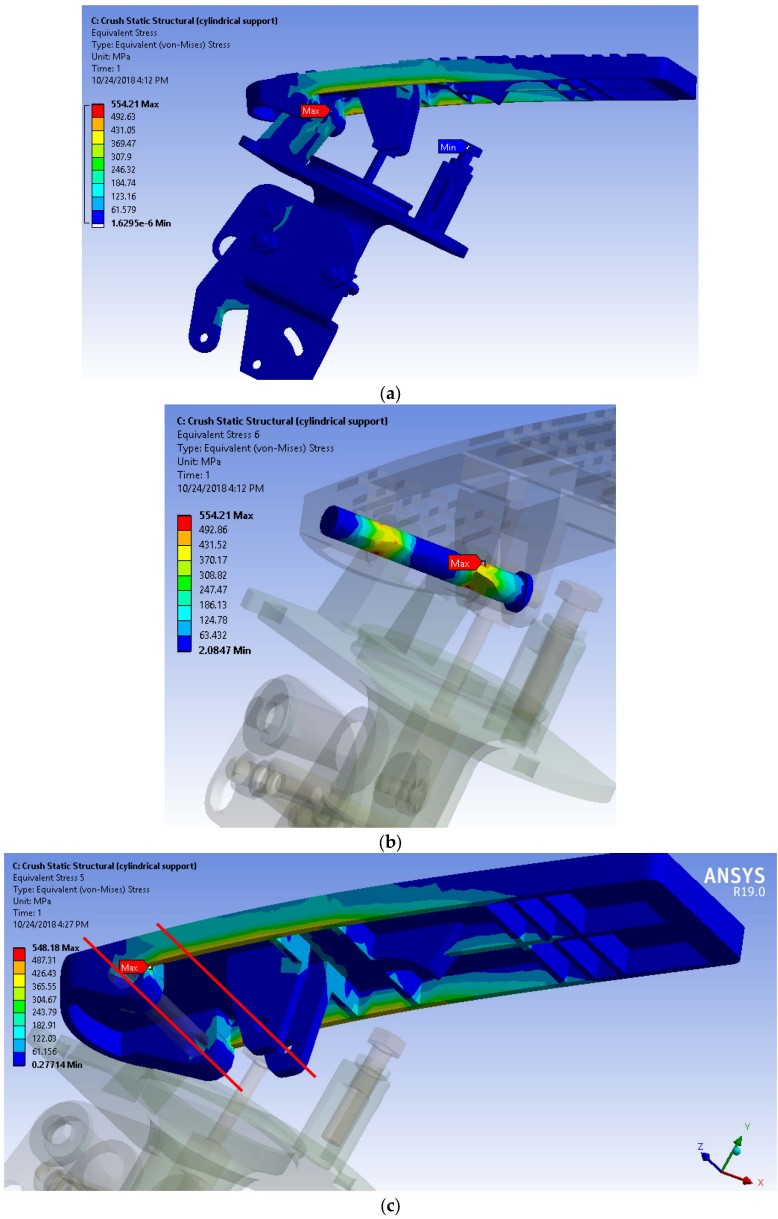

**Figure 12.** Stress maps of the pedal model under the hybrid task: (**a**) assembly; (**b**) pin; (**c**) foot part of pedal.

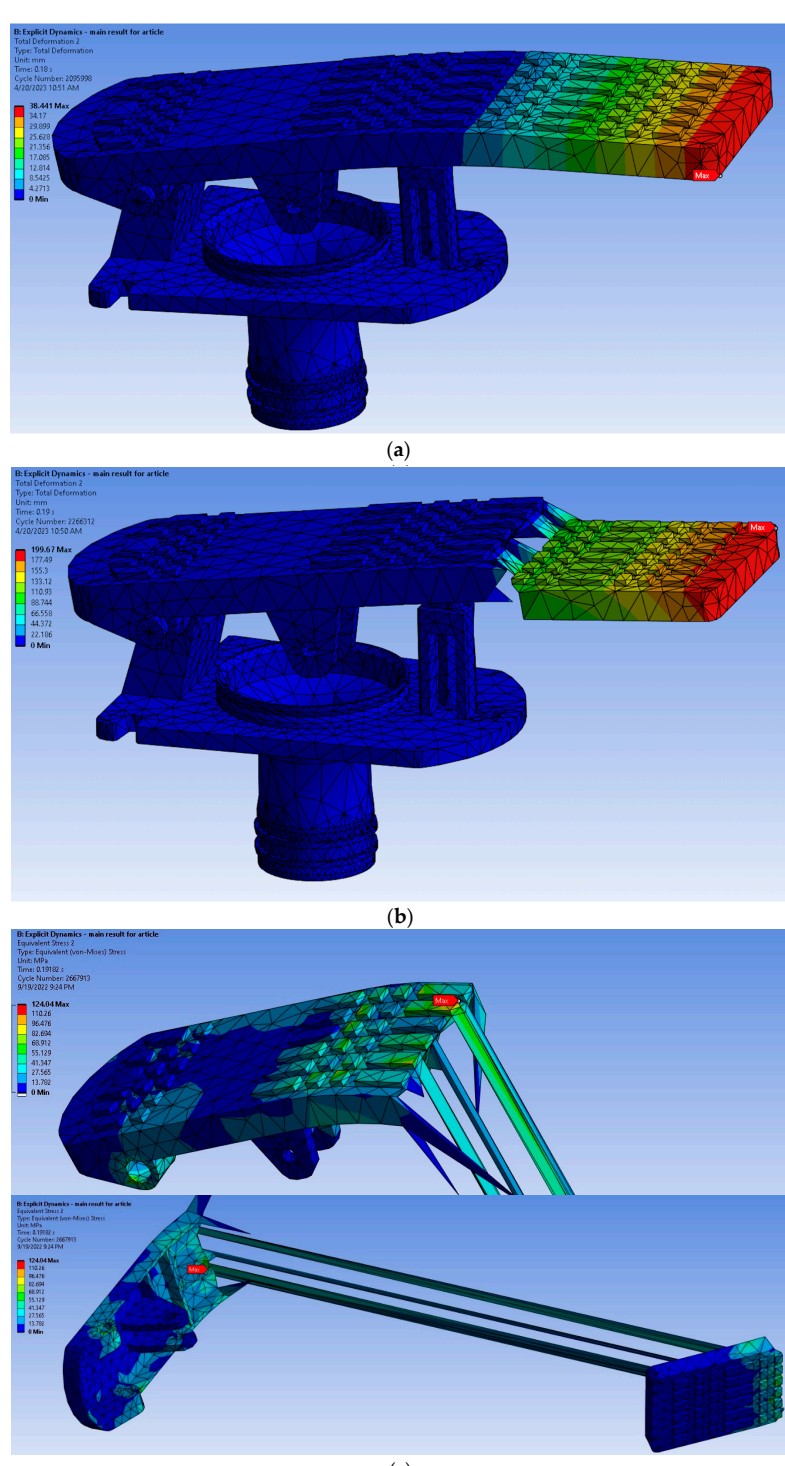

**Figure 13.** Pedal stress in conditions of shock loads task—views of stress maps: (**a**) before the rupture; (**b**) first moment of rupture; (**c**) corrupted pedal.

**Table 3.** Comparison of calculated result and average result of natural experiments.

| Pedal Breaking Stress, MPa | | Pedal Breaking Load, N | | |
|---|---|---|---|---|
| Yield Stress of Silumin | Calculated | Experimental | Calculated | Error Δ, % |
| 120 | 124 | 2255.5 | 2500 | 9.8% |

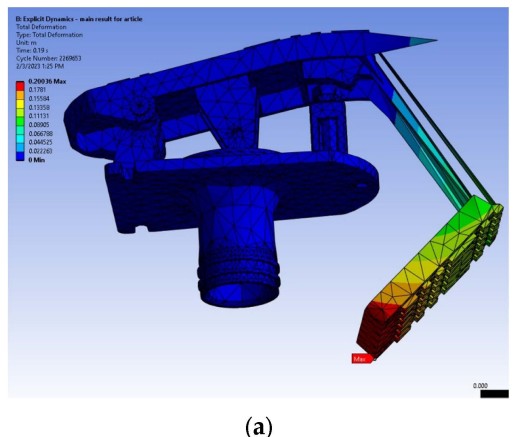

(**a**)

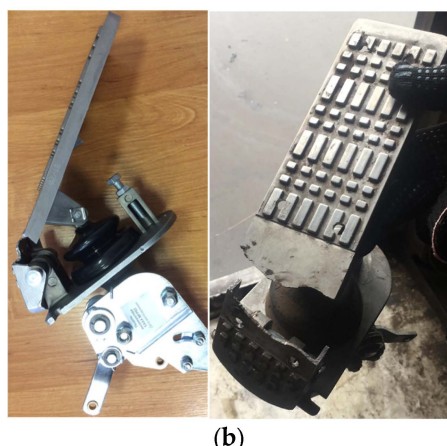

(**b**)

**Figure 14.** Corrupted pedal: (**a**) stress in conditions of shock loads task: total deformation; (**b**) photos of the destroyed samples.

## 4. Experimental Test

Apart from the calculations carried out in Ansys Explicit Dynamics, the experimental tests were arranged as well. The principal scheme of the experimental setup (Figure 15) explains its mechanism of work.

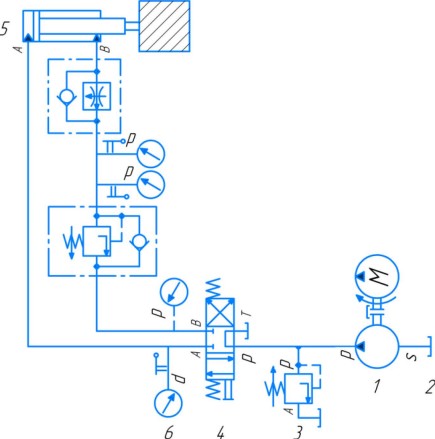

**Figure 15.** Principal scheme of the test stand for checking the strength of the pedal (by Authors).

Pump 1 sucked hydraulic fluid from reservoir 2 and delivered it to the hydraulic system. The flow distributor with manual control 4 was in the neutral position. Its valve was held in the neutral position by two centering springs. When the distributor 4 was turned on (left position, indicated by parallel arrows in Figure 15), the working fluid entered the cavity of hydraulic cylinder 5, and the piston rod extended. The force exerted on the piston rod depended on the area of the piston itself and the maximum pressure in the hydraulic system. The maximum pressure in the hydraulic system, and therefore its load, was controlled by pressure-limiting valve 3. The value of the pressure in the system was determined by the pedal's resistance being overcome and was measured by manometer 6.

The real test stand (Figure 16) consisted of a stationary base on which a hydraulic station (oil tank, high pressure hydraulic pump and cylinder, hydraulic distribution equipment) and the platform for mounting the pedal were installed. The pressure in the hydraulic cylinder was created by a pump and was regulated by the corresponding valves (red ones in Figure 16). Being set in the extreme position, the pedal rested against the locking bolt, which served as a limiter for its travel (this corresponded to the position of pedal in Figure 14). Thanks to the hydraulic valve system, it was possible to adjust the pressure and speed of the rod's travel accordingly. To control the indicators, dial gauges were used to

determine the pressure in the hydraulic cylinder (manometer), as well as the force of the rod (dynamometers). The operator monitored the load indicators and wrote critical results (the point at which the pedal was destroyed) in a journal. The initial load moment (when the rod started pushing the pedal) is demonstrated in Figure 16a. In a moment when the pressure increased and the pedal continued to deform, the stress reached a value above the yield point and the pedal cracked (final moment—Figure 16b).

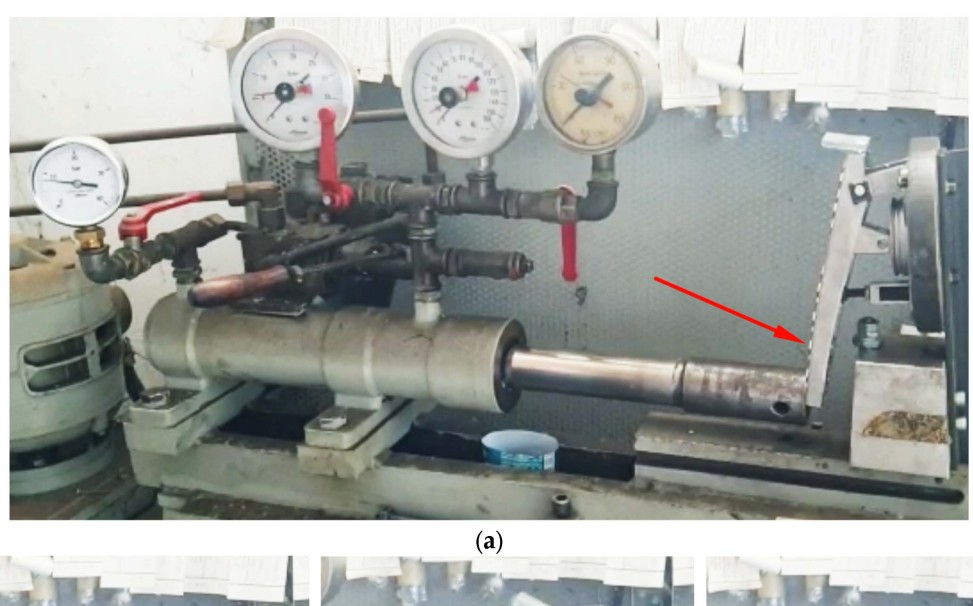

(**a**)

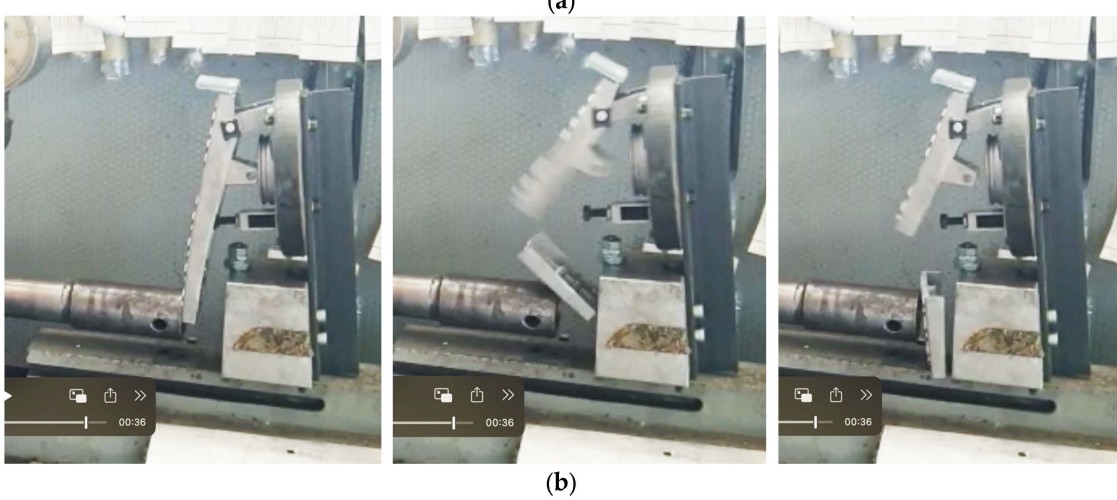

(**b**)

**Figure 16.** Full-scale strength tests of EAAH-MFP02 pedal: (**a**) moment of testing before breaking (initial load time instant; a red arrow points the pedal); (**b**) breaking point.

The obtained results (Figures 13 and 14) of the shock load task are close to the results of the full-scale experiments (Table 3), which have demonstrated the load of 230 kg (2255.5 N) on the rod. This indicates the sufficient effectiveness of the presented methodology for defining the pedal strength, taking into account the comparatively high convergence of simulation results and full-scale tests (9.8%).

Apart from the breaking tests described above, it is worth explaining that, in fact, each sample of the pedal (model EAAH-MFP02-2340) has to pass the load tests individually, and is checked before being forwarded to the client (tractor factory); therefore, the manufacturer of these pedals needs to perform a qualification check each time. The tests are performed for assembled units, which consist of the pedal and the push–pull cable mounted on the stand (Figure 17a). Unlike the previously described stand (Figure 16), where the pedal rested against the locking bolt serving as a limiter for its travel, this stand limits the pedal's travel with the help of the push–pull cable, whose end is connected to the strain gauge,

which, in turn, is connected to the digital dynamometer (Figure 17b). Therefore, the pedal is mounted to the stand, and the piston rod pushes on its foot part (Figure 17a); the cable accepts the load from the pedal and tries to transfer it to its another end, which is connected to dynamometer (Figure 17b). Therefore, the operator can check the transfer of loads from the pedal to that other end of the cable (there should be no loss of cable travel due to twisting of the core inside the housing). As described in "Hybrid task" (Figure 7b), the input load applied on the pedal was twice lower than the output load on the push–pull cable, i.e., the load transfer coefficient for the pedal was equal to 2. The main goal of this experimental stand was to confirm whether the assembly (pedal with cable) followed the manufacturer's specifications regarding its safety margin and reliability, as well as guaranteeing the transfer of the necessary load by the cable.

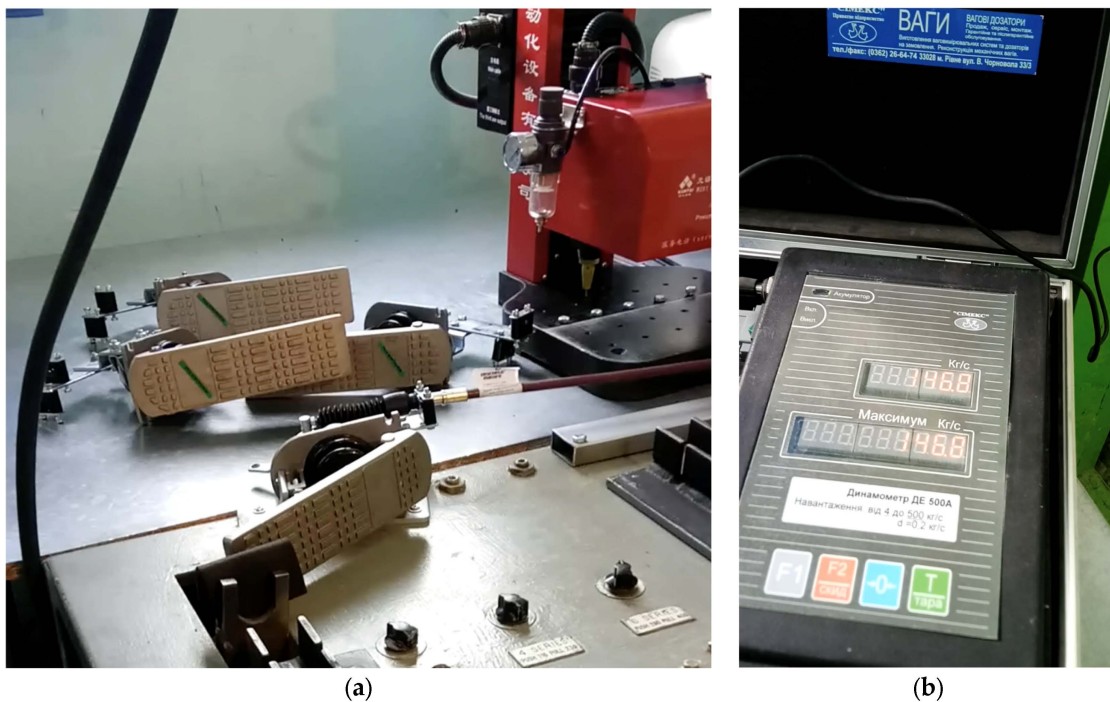

(**a**)                                                                                  (**b**)

**Figure 17.** Experimental stand for checking of the cable load transfer losses and loads: (**a**) pedals used in the tests; (**b**) the digital dynamometer presenting results of the experiment.

## 5. Discussion

Unlike UNECE R 13 [1] described above, the requirements of UNECE R 35 [3] concern the ergonomics of the driver's workplace, in compliance with the demands for a comfort and the efficiency of the control elements' interactions with the driver, which has been addressed in [4–7]. As was demonstrated, the topic of comfort is not enough when it comes to the safety of braking based on the driver's reaction and the speed of the pedal activation and the pedal strength, as is presented in [9–13,15,18,19]. The main issue is that all those publications were dedicated to the brake pedal strength, but none of them applied to an evaluation of the accelerator pedal strength under critical loads, which is quite important in real operational conditions.

Thanks to the availability of the equipment (test benches) presented above, it is possible to check not only the load causing mechanical destruction of the pedal (Explicit Dynamics calculation case), but also to measure the load in the process of pressing the pedal and transferring forces to the executive unit through the remote control cable (Transient Structural calculation case). This approach allows modeling intermediate loading processes and the final failure moment (over 1000 N of force). The study of the intermediate states of the pedal is important for the practical operation of the tractor (model K 744R); in fact, during its physical tests by the manufacturer, the executive unit jammed several times, so

the remote control cable could not transmit the load from the pedal. The reaction from the jammed unit was returned to the pedal through the push–pull cable, which was simulated in the hybrid task conditions. A real pedal's strength varies within ca. 8–12%, depending on a sample's lot. This is explained by the value of the actual strength limit of Silumin 4000, which in turn depends not only on the chemical composition of the alloy, but also on the casting technology (pressure casting). That is why multiple samples of the pedal were tested for strength within the experiments. It could be suggested to apply an ultimate strength condition based on the yield strength of Silumin, taking into account the too-short period of plastic deformation before the pedal breaks; this was found in the real-life experiments (Figure 16b) and FEA simulation (Figure 13a,b and graph in Figure 8). In other words, a zone after the yield point became too risky in the presented research to judge the safety of the pedal based on its tensile strength, which significantly depends on the method of pedal casting used, and varies quite a bit within the samples.

Another limiting factor is the remote control cable (push–pull cable) itself—according to the technical documentation, the load it must transmit should not exceed 100 kg. A dilemma arises here: what should break first? If the cable has an excessive safety margin, the victim will be the pedal, which is a reason why it is advisable to simulate load processes and physically test the pedal assembly with the cable. In general, the value of the 100 kg limit of transmitted forces on the cable, and thus 1000 N on the pedal, is not accidental and comes from the requirements of UNECE R 13. At the same time, it has been observed over a double safety factor gap in the case of the pedal: the force has reached 2255 N.

The uniqueness of the presented research lies in the fact that the UNECE requirements regarding the strength of the brake pedal have been transferred to the accelerator pedal, and boundary conditions of the FEA calculation equivalent to full-scale ones have been formed, and this was proven by real tests. Here, the publications by Hfaiedh et al. [14], Lee et al. [16], Balakrishnan et al. [20], and Khandani [21] can be helpful: they are dedicated to explicit calculations in the finite element method, which was also used in the presented study. This approach can serve as a basis for the formulation of other individual methods of testing pedals (clutch, other units).

One of the ideas which arose at the end of the provided investigation is to equip pedals with a tensor strain gauge, which would break off in case the load exceeds 1000 N—it will signal that the executive unit jammed or the push–pull cable route is wrong (too-small radii or too-curved route causing a jamming of the cable). An additional advantage of such a tensor strain gauge on the pedal is that it can demonstrate that the pedal has successfully passed the loading tests.

## 6. Conclusions

Results presented in the current research allow us to draw final conclusions about the applicability of simulation tests using the FEA method in the Ansys environment (Transient Structural and Explicit Dynamics) to assess the strength of a mechanical pedal in various operating conditions.

1.  The obtained concordance of results between experimental and simulated tests (less than 10%) indicates the sufficient economic efficiency of the pedal behavior research methodology proposed in the work.
2.  The studied method will be useful for design studios, where engineers are involved in the development of new pedals, and in workplace layout investigations. At the same time, the theoretical knowledge presented in the article regarding the explicit and implicit approaches in FEA leads to the idea that the explicit-type calculations used in Ansys are very resource-intensive: they require a setting of a very small time step and a huge number of iterations; thus, the calculation requires a lot of hours on sufficiently powerful equipment. For this reason, an optimization of the structure and strength of the pedal in the Ansys Transient Structural environment is first recommended, and then the use of the Explicit Dynamic environment just at the final stage, for crash tests.

3. The lack of regulatory requirements for the strength of pedal types other than brake pedals is a major gap in vehicle certification, especially when it comes to agricultural machinery (such as tractor K 744R, as in the presented study). In such cases, the authors suggest being guided by UNECE R 13 regarding the strength of the accelerator and other types of pedals, and checking their behavior under loads of at least 1000 N. The studied pedal, EAAH-MFP02-2340, used on tractor K 744R, withstood over twice the standard load (2255 N), which coincided with experimental studies within a 10% margin of error.

4. For a comprehensive assessment of the pedal's strength, its behavior must be modeled in complex, taking into account the simulation of the cable and the corresponding executive unit along with the application of reactions and other boundary conditions. Determining the failure loads is not a sufficient result; it is also necessary to check the intermediate stress values depending on the loads. For this, it is advisable to carry out calculations for direct and hybrid tasks using the Transient Structural module. The Explicit Dynamics module should be used for the simulation of pedal rupture (stress was 124 MPa, which exceeded the yield point).

5. The real meaning of the yield strength of the material (Silumin 4000, in our case) is very important, both in the physical real-life experiments and in the FEA imitation. The way the casting was produced (free cast or under high pressure) significantly influences the total strength of the pedal and its safety margin; hence, it could be suggested to apply an ultimate strength condition based on yield strength and to install a tensor strain gauge on the pedal, which would break off in case the load exceeds 1000 N. This strain gauge could be recommended to be installed on each pedal and its investigation could be the topic of further research.

**Author Contributions:** Conceptualization: K.H., O.H., E.K. and I.K.; methodology: K.H., O.H. and S.B.; validation: O.B., V.P.; investigation: K.H.; data curation: V.P. and R.H.; writing: K.H. and O.H.; writing—review and editing: K.H., O.H. and M.C.; visualization: K.H., O.B. and A.L.; supervision: K.H., V.P., O.B. and M.C. All authors have read and agreed to the published version of the manuscript.

**Funding:** This research received no external funding.

**Institutional Review Board Statement:** Not applicable.

**Informed Consent Statement:** Not applicable.

**Data Availability Statement:** Since no new data were generated or examined in this study, data sharing does not apply to this article.

**Acknowledgments:** We would like to express our gratitude to Mykhailo Burian and Yaroslav Sholudko for their active help in a creation of this paper.

**Conflicts of Interest:** The authors declare no conflict of interest.

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
