# Peer review of "Evaluation of Accelerator Pedal Strength under Critical Loads Using the Finite Element Method"

_applsci, doi:10.3390/app13116684_

Round 1

Reviewer 1 Report

1. In the Introduction section, it is necessary to emphasize the relevance of the study. At the moment, this section is quite vague.

2. The analysis of literary sources and publications needs a complete revision. It is written in a very concise manner and contains little to no critical analysis. This section should show the previously unsolved problem and end with the argumentation of the need for research in this direction.

3. The authors need to highlight the purpose and objectives of the study.

4. In the “Materials and Methods” section, it is necessary to enter information about the geometric characteristics of the object of study, the features of creating its spatial model, the assumptions that were made, etc.

5. There is no information about the features of creating a finite element model, namely, how the number of elements was determined, checking the sensitivity of the model, etc.

6. The equations given in the article are well known. It is not clear the need to bring them.

7. Fig. 8 requires clarification. How was it obtained?

8. The article lacks verification of the adequacy of the results obtained. The authors compared the results of computer simulation with a physical experiment and compared the results obtained for a single point. It is advisable to obtain a spectrum (sample) of values by computer simulation and a physical experiment with subsequent verification.

9. The article does not contain the section “Discussions”, which should contain information: how the results of the study can be explained, due to what features the advantages of the study are provided compared to the known ones. Next, it is necessary to indicate how the results obtained close the problematic part of the study. Indicate the limitations and shortcomings of the study, as well as its future prospects, indicating why in this particular way.

10. The conclusions will need to be structured in accordance with the objectives of the study.

11. The list of literary sources also needs to be expanded. To publish an article in such a journal, the number of sources is very limited

Reviewer 2 Report

see the attached

Reviewer 3 Report

In the article "Evaluation of accelerator pedal strength under critical loads using the finite element method", authors studied stress, strain and safety factor of accelerator pedal of varying loads and boundary conditions. Authors argued that the proposed method of analysis can be considered as universal for assessment of the strength of typical vehicle pedals. After careful review of the article, I have concluded that the article needs major revision and can't be accepted for publication in the current form. Few comments to the authors are given below:

1. poorly stated abstract. Does not provide the problem statement, needs, findings, and recommendations.

2. very poorly written introduction section. literature review is inadequate, references are somewhat irrelevant. Ref 12 and 13 is the only relevant references, however there is not insight from those articles. Authors did not discuss gaps, results or findings from those ref. Therefore, readers won’t have any idea of the novelty of this research.

3. Fig1 has no implication and poorly discussed. Why there is an image of destroyed sample? A detailed model can be added with proper annotations for readers to understand the overall structure under study, its mechanism and load path, critical region, critical condition of failure, etc.

4. fig2(b) shows finite element mesh of the model which clearly shows very poor coarse mesh with only 17881 element. In the second paragraph of section 2.1, authors said “the parameters meeting the accuracy requirements of Transient Structural module regarding the convergence of results”. The convergence of numerical solution and accuracy of the results are not same. Authors did not explicitly mention how they achieved that. In any article with FEM results it is customary to show mesh convergence study by tracking critical parameters and their relative percentage error.

5. Authors used transient model in Direct Task. Clearly the load is applied very slowly compared to the dynamics of the system (fig 3). Authors can use Eq2 and justify why transient loading case is not necessary if the inertial term can be neglected or they can show that transient case is absolutely necessary because it gives more insight of the system. If we can’t draw any more information using transient model that we really don’t need it. A quasistatic analysis with multiple loadsteps is sufficient perhaps.

6. in section “2.1 Direct Task” authors have mentioned, to some extend, how loads/boundary conditions are applied, but pedal, base, vehicle floor all those features are not shown in fig4 except few red arrows. An article should be standalone, means readers should get all necessary information from the article itself. Authors should not assume that readers know the pedal system in detail.  

7. overall, all figures need to be improved and more professional scientific practice need to be followed from relevant research articles (e.g., which one is the extreme position in fig. 9a?).

8. section 2.3 Shock load task has 12 basic equations. Why these equations are needed? Is there any contribution from the authors to these equations? If not, can they be just referred to any textbook or ansys documentations? Why we need fig 8 in this article? What is the take away information from fig 8?

9. Results sections need improvement. It is absolutely clear from fig 13 and 14 that proper failure criteria is not provided. There are extremely distorted elements. Authors have said the stress is 124MPa (fig 13) which exceeded the yield point. From table 1 it seems the yield varies from 120-180MPa. Which one is used? the lowest value (120MPa)? Why? If authors have used 180MPa what would happen? What would be the failure load then? In explicit dynamics simulations, how the nonlinearity of the material is included? From table 1 the difference between tensile strength and yield strength shows the material behave as ductile material. It should go through plastic deformation after yielding. How this section of the material properties were added in ansys? I assume it is input as isotropic materials (since there is no explanation from the authors) with only Possion’s ration and young’s modulus. In this case the material in FEM simulation can have infinite stress and strain. They will just linearly increase. Probably that’s why there are lots of highly distorted elements which are having extreme strain after yield point.    

10. is the section 4. Experimental test added to validate the results from FEM simulations? In that case table 2 needs significant improvements. Did authors test the specimen just once? If yes, as there is only one value in table 2, then the test result may not be acceptable. Please include a repeated test results and explain how reproducible your results are, what are the variations in the test data (including uncertainty, error), what are the confidence level etc.

Reviewer 4 Report

This paper presents a practical application of FEM in the development of an automotive design element. The validation of the computational approach with experiment is interesting and important. The paper contains a significant amount of work in the field of innovative engineering. The paper is certainly suitable for publication in Applied Sciences, but I give consideration to whether a greater impact would be made by publishing it, for example, in the journal Vehicles - MDPI.

Author Response

Thank you very much for your kind feedback. We are honored to accept this.

Reviewer 5 Report

The quality of editing and English language is very good, the explanations are clear.

It is a restraint case study, the interest of publishing it in a scientific journal is to be discussed. The real public would be in a relevant conference.

Author Response

(The authors gave the same response as above.)

Round 2

Reviewer 1 Report

The authors listened to some of the comments. However, in my opinion, the following questions remain unanswered:

- there are no research tasks;

- there is no verification of the obtained results. What is stipulated in the Discussion is not verification.

- the well-known formulas of the finite element method, which are present in textbooks, are superfluous in an article for such a publication. In addition, the authors did not even make a reference to the literary source from where they were taken.
